# Atmospheric Plasma Treatment to Improve PHB Coatings on 316L Stainless Steel

**DOI:** 10.3390/polym16142073

**Published:** 2024-07-20

**Authors:** J. Radilla, H. Martínez, O. Vázquez, B. Campillo

**Affiliations:** 1Laboratory Espectroscopia, Instituto de Ciencias Físicas, Universidad Nacional Autónoma de México, Mexico City 04510, Mexico; hm@icf.unam; 2Laboratory Metal Mecánica, Instituto Tecnológico de Morelia, Morelia 58120, Mexico; octavio.vg@morelia.tecnm.mx; 3Facultad de Química, Universidad Nacional Autónoma de México, Mexico City 04510, Mexico; bci@icf.unam.mx

**Keywords:** atmospheric plasma, PHB, coating

## Abstract

In the present study, biopolymeric coatings of polyhydroxybutyrate (PHB) were deposited on 316L stainless steel substrates. The PHB coatings were developed using the spin coating method. To improve the adhesion of the PHB coating on the substrate, this method uses an atmospheric plasma treatment. Adhesion tests show a 156% increase in adhesion after 5 s of surface treatment. Raman spectroscopy analysis of the polymer shows the incorporation of functional groups and the formation of new hydrogen bonds, which can help us bind drugs and promote osteogenesis after plasma treatment. Additionally, the electrochemical behaviors in artificial body fluids (Hanks’ solution) of the PHB coatings on the steel were evaluated with potentiodynamic tests, which revealed a decrease in the corrosion current and resistance to the transfer of the charge from the electrolyte to the 316L steel because of the PHB coating. All the PHB coatings were characterized using scanning electron microscopy and Raman spectroscopy after the electrochemical tests. This analysis confirmed the diffusion of electrolyte species toward the surface and the degradation of the polymer chain for the first 15 s of treatment with atmospheric plasma. These findings support the claim that plasma surface modification is a quick, environmentally friendly, and cost-effective method to enhance the performance of PHB coatings on 316L stainless steel for medical devices.

## 1. Introduction

Implants in the context of biomaterials are constructs that have instructive/inductive or triggering/stimulating effects on cells and tissues. The materials are engineered to respond to internal or external stimuli and have intelligently tailored properties and functions that can promote tissue repair and regeneration [1].

316L stainless steel, MgAl alloys, and CoCr alloys are used to make traditional metallic stents and medical devices such as screws and plates. Additionally, 316L stainless steels (SSs) are of great importance in the aerospace, marine, gas, and chemical industries as they have a high Cr-Ni content and a Cr element content between 16% and 18% [2,3,4]. However, after prolonged contact with human tissue (including corporal fluids), corrosion occurs on the surface of these alloys, resulting in the unwanted release of transition metal ions, such as chromium, iron, and magnesium. This process can lead to musculoskeletal injuries, including bone defects due to suboptimal healing [5,6,7]. 

Polyhydroxyalkanoates (PHAs) are a group of biodegradable polyesters produced by bacteria over the carbon source and a potential solution to the abovementioned issues. Poly(3-hydroxybutyrate) (PHB) is the best-known polymer of PHAs. Its properties, biodegradability, biocompatibility, and non-cytotoxic metabolic products have led to its widespread use in biomedicine and tissue engineering. Over the years, biodegradable polymer scaffolds have been used to regenerate and replace soft and hard tissues [8,9,10,11]. 

The widespread application of biopolymers in this sector offers significant advantages. Forming coatings on the substrate reduces the magnesium degradation rate by creating a microenvironment with a local pH value of 7.4 to 8.8 and an ion concentration favorable for osteoblast proliferation while preventing the release of metal ions. Results indicate that coating alloys with the biodegradable polymer PHB improves corrosion prevention compared to uncoated alloys [11,12,13].

Some polymers are used for the effective identification of alkaline/acidic vapors, which is indispensable for detecting toxic substances in products used in industrial production or for everyday needs. These substances, such as acetic acid and ammonia vapors, are deceptively hazardous to human health. Additionally, research has been conducted to manufacture PVC-based compounds using biocarriers with acceptable performance characteristics [14,15].

Present efforts focus on improving the physicochemical properties of biomaterials and implants to promote interactions with the host tissue and osteogenesis. Biomaterials are designed to actively interact with stem/progenitor cells and the extracellular matrix (ECM) to influence the local environment toward osteogenesis and tissue regeneration [5].

Numerous methods are used to fabricate materials and modify surfaces; however, non-thermal atmospheric pressure plasma (APP) jets are widely used due to their simple design, low power requirements, environmentally friendly nature, and selective characteristics related to plasma production. Critical operating parameters such as pressure, discharge type, gas type, flow rate, substrate temperature, and distance are the variables that affect polymer properties [16,17,18].

As a result, in the present work, we studied biopolymeric coatings of PHB on SS to improve their adhesion with APP. This marks the first step toward the study, creation, and development of functional medical implants that are not rejected by the human body.

## 2. Materials and Methods

### 2.1. Steel and Polymer Preparation

The SS was purchased from the Morelos steel and metals distributor in the form of a 5.04-inch bar. Fifteen discs, each with a diameter of 4 cm and thickness of 5 mm, were machined, and all samples were roughed with “silicon carbide (SiC)” sandpaper as an abrasive. Grain sizes ranging from 300 to 1200 were used to generate a homogeneous surface in all the tests. The pieces were washed and cleaned of impurities by placing them in an ultrasonic vibrator with acetone for 30 min. They were then dried in an oven at a temperature of 35 °C for 1 h. The polymer used was purchased from Merck company (linear formula: [COCH_2_CH(CH_3_)O]n, CAS No: 29435-48-1). Then, 1.8 g of PHB was weighed for every 50 mL of chloroform. The dissolution of PHB to form the coatings was carried out in an electric grill with magnetic stirring under the following conditions: T = 60 °C, t = 2 h, and a 400 (rpm) stirring speed. The PHB solution (3 mL for each sample) was injected onto the SS at 300 rpm for 1 min using a spin coater. The thickness of the coatings was measured in triplicate using a WEN brand digital micrometer (WEN Products, Elgin, IL, USA). The film thickness obtained was 15 ± 1 μm.

### 2.2. Treatment with Atmospheric Plasma and Coating Formation

To determine the best procedure to coat SS with PHB using APP, we used the formula for combining 3 elements (SS, PHB, APP) to get the best coating, that is $C_{1}^{3}=\frac{3¡}{1¡2¡}=3$. That gives three different methodologies for the formation of the coatings SS-treatment/coating/coating treatment (M1), SS treatment/coating (M2), and SS/coating/coating treatment (SS-PHB), each with different treatment times. The coatings were developed using the spin coating method. The methods were assigned the abbreviations M1, M2, and SS-PHB for practical purposes. The treatment times used in this investigation are shown in Table 1. All treatments were performed with the plasma outlet positioned 4 cm from the sample’s surface. The treatments were conducted using APC500 Spray Corona atmospheric plasma equipment from Diener Electronic, Ebhausen, Germany, operated at a voltage of 20 kV, a discharge current of 25 mA, and a frequency of 40 kHz. With the present condition, the electric field strength (*E = V/d*) was 5.0 × 10^3^ V cm^−1^, E/*P* = 6.58 V cm^−1^-torr^−1^, an electron drift *v_d_* = 3.2 × 10^6^ cm-s^−1^, and *E*/*N* of 0.19 *Td*. That gives an electron density of (1.95 ± 0.24) × 10^12^ cm^−3^ [19]. The APP was also characterized by optical emission spectroscopy from 200 to 900 nm; the lines and bands observed were OH (A^2^Σ^+^ − X^2^Π_i_), N_2_(C^3^Π_u_ → B^3^Π_g_), N_2_^+^(B_2_Σ_u_^+^ → X^2^Σ_g_^+^), O (^5^S^0^ − ^5^P), and O (^3^S^0^ − ^3^P). The electron energy of the plasma was found using an optical characterization of the discharge, from the ratios of the O atomic line intensities in the emission spectrum. The electron energy obtained was (0.36 ± 0.04) eV [19].

### 2.3. Contact Angle (CA) and Surface Free Energy

The wettability of the steel and the coatings treated with APP was evaluated using the sessile drop technique to calculate the surface free energy. The volumes used were 3 µL of diiodomethane and 5 µL of distilled water. Three measurements were made with each fluid for each sample. The images were recorded with a ‘MicroView’ 1000× digital microscope (MicroView, Lagos, Nigeria), and the contact angle values were obtained from the geometric analysis of the droplet images using the ‘ImageJ’ version 2.9.0.

We used the Owens–Wendt mathematical model and the method described in reference to obtain polar and dispersive components as well as surface free energy [19]. Thus, the model allows the calculation of the surface free energy of the solid using the contact angle between the drop and a surface. The components of the surface tensions of the standard liquids used were taken from reference [19]. The fluids used in this study were distilled water and diiodomethane with 99% purity supplied by Sigma-Aldrich (St. Louis, MO, USA).

### 2.4. Pull-Off Adhesion Test

Three samples were prepared for each treatment time for the M1, M2, and SS-PHB methodologies. Adhesion tests were performed according to ASTM D4541, which measures the force required to detach a coating from a metal surface of a specific diameter from the substrate using hydraulic pressure. The pressure was calculated based on the force and area of the test, representing the adhesion force. The equipment used was the Defelsko brand (Ogdensburg, NY, USA) and the manually operated “PosiTest AT-M” model.

### 2.5. X-ray Diffraction (XRD)

All samples with and without APP treatment were analyzed using a “Rigaku Miniflex DMAX 2200” X-ray diffractometer (Rigaku Miniflex) in Austin, TX, USA, to evaluate the changes in crystallinity due to the interaction with the APP on the PHB coating. The instrument was equipped with a Cu Kα (1.54 Å) source and a graphite monochromator operating in the 2θ range of 10–60 with a grazing angle beam. The percentage of crystallinity (%C) was calculated using Equation (1), which considers the area under the first five diffracted peaks and the total area under the diffractogram curve [20].
(1)$$\%C=\left(\frac{I_{c}}{I_{c}+kI_{a}}\right)\times 100$$
where %C represents the crystalline fraction in the analyzed zone, which is representative of the area studied using the spectrometer. Ic is the area under the curve of the first 5 diffraction peaks at an angle of 2θ in the analyzed diffractogram, and Ia is the area under the curve obtained by integrating the 2θ diffractogram from 130 to 260. Additionally, k is the constant of characteristic proportionality of the PHB polymer.

In addition, the sizes of the crystallites formed by the crystalline structure of the PHB were calculated using the Scherrer equation (Equation (2)) [21,22]:t = 0.9 λ/(ß cos θ_B_) (2)
where λ is the wavelength of the X-ray, in this case, 1.24 A Å (wavelength of Cu); ß is the width of the peaks (in radians) where the intensity value corresponds to half the maximum intensity; and θ_B_ is the angle, in radians, at which the intensity is maximum. The (021) plane was used for the calculated crystallite size.

### 2.6. Characterization with Optical Raman Microscope

A SENTERRA II (Bruker) Olympus Raman microscope (20× objective) was used to obtain a qualitative evaluation of the PHB coating treated with APP.

### 2.7. Raman Spectroscopy

Raman spectroscopy chemical analysis was used to study the symmetric and antisymmetric modes of rotation and vibration of the compounds that form the PHB polymer chain before and after APP treatments, as well as after electrochemical tests. We aimed to determine whether functionalization of the surface of the PHB coating occurred, as this can help promote the adhesion of nanoparticles and drugs. The degradation of the PHB coatings was also assessed using a “SENTERRA II (Bruker)” Olympus (20× objective) Raman microscope with OPUS 7.8 software, a laser with a wavelength of 785 nm, and a power of 100 mW

### 2.8. Polarization Curves and Electrochemical Impedance Spectroscopy

The electrochemical tests were conducted using a three-electrode cell in the “Corrtest model CS350” potentiostat (Corrtest, Wuhan, China). The setup included a saturated silver chloride reference electrode, a platinum counter electrode, and a working electrode made from the samples under study. The medium was 250 mL of Hanks’ solution (composition: 8.0 g/L NaCl, 1.0 g/L glucose (C_6_H_6_O_6_), 0.4 g/L KCl, 0.35 g/L NaHCO_3_, 0.14 g/L CaCl_2_, 0.06 g/L KH_2_PO_4_, 0.098 g/L MgSO_4_.7H_2_O, and 0.048 g/L Na_2_HPO_4_). The solution was kept at a constant temperature of 37 °C, and the electrochemical cell was heated and placed on an electric rack without stirring. The tests began with an open circuit potential measurement for approximately 3000 s, followed by electrochemical impedance spectroscopy in an area of 0.785 cm^2^. Polarization curves were evaluated in the same area, with a sweep speed of 1 mV/s and a voltage range of −1 V from 2 V at a frequency of 2 Hz.

## 3. Results

### 3.1. Pull-Off Test

Figure 1 shows the evaluated results of the adhesion tests of the PHB coating on SS with APP for the three different coating formation and surface treatments methodologies. In Figure 1a, the time shown on the *x*-axis (min-s) refers to the SS treatment time (min) and the treatment of the PHB coating (s) for M1. In the samples of M1 (Figure 1a) and M2 (Figure 1b), a decrease of approximately 63% and 40% in the adhesion of the PHB coating is observed after 5 min of treatment. This may be due to an increase in the surface free energy of the SS and an increase in the polar and dispersive contributions, which can influence the decrease in the adhesion of the PHB coating on the SS. After 5 min and up to 15 min, the strain follows a visible stabilization trend in methodologies M1 and M2 due to a stabilization of the polar component in the SS, which is confirmed in the next section. These results may indicate a relationship between the adhesion stabilization and the stability of the existing polar components on the SS surface because the variation in the measurements in the polar component is insignificant. The increase in the dispersive component does not affect the stability of the strain behavior. Observations show that, for 20 s, the M2 sample presents an increase in the strain of approximately of 36%, which can be correlated with an increase in the dispersive component.

Regarding SS-PHB, it is shown that the treatment of the PHB coatings with APP increases the strain by 55% in the first 5 s of treatment, followed by a decrease of 17% at 10s, and finally, between 10 and 20 s of treatment, an increase in the strain of 69% is observed with respect to the untreated sample. This behavior may be attributed to the surface free energy of the PHB coating, which favors adhesion at the SS-PHB polymer interface, contrary to the increase in free energy in SS.

Since SS-PHB was found to be the best methodology for obtaining the optimal PHB coating, all subsequent analyses were conducted using this approach.

### 3.2. Contact Angle (CA) and Surface Free Energy (SFE) of the Treated Stainless Steel (SS)

As shown in Figure 2a,b, the SFE for untreated SS was measured as 39.20 mJ/cm^2^, which yields contact angles greater than 50° for both fluids, with low contributions from the polar component. Some molecules contributing to this process are oxides and hydroxides. SFE is the sum of polar and dispersive contributions related to intermolecular forces formed on the surface. The polar contribution arises from hydrogen bond formation, while the dispersive component results from London forces without hydrogen bonds, meaning that electrostatic forces of attraction and repulsion interact between different phases [23].

In the sample treated at 5 min, contact angles for both fluids decreased, resulting in more significant contributions from each component and an SFE of 65.39 mJ/mm^2^. This indicates a chemical change on the surface of stainless steel treated with APP due entirely to particles in the environment ionized in the plasma and introduced to the SS surface.

At treatment times of 10 min and 15 min, the increase in SFE of the SS is not very significant compared to that at treatment times from 0 to 5 min. This could indicate an activation energy barrier of atoms functionalizing the surface or stabilization of particles due to a limit of particles introduced to the SS surface being reached. However, the SFE still increases, which may be related to electronic density distributed on the SS surface, as this can generate instantaneously induced dipoles.

From 5 min to 20 min, the contribution of the polar component remains slight, and the component that most influences the SFE is the dispersive component. The sample treated at 20 min presents an energy of 75.43 mJ/mm^2^, indicating that significant chemical changes occur on the SS surface due to the interaction of the particles formed in the environment ionized in the plasma that interact with the SS surface. This results in an increase in SFE after 5 min of treatment with APP. The molecules of diiodomethane and water experience a change in the polarizability of their atoms upon contact with the surface containing oxide and hydroxide molecules, which produces a change in the contact angle of both fluids with the SS surface.

### 3.3. Contact Angle (CA) and SFE of the Coating

In Figure 3a,b, the contact angles and the SFE are shown as a function of the APP treatment time of the PHB coating. The untreated sample has a hydrophobic tendency, evidenced by the contact angle being greater than 90° for distilled water, while diiodomethane has a contact angle of 17.9°. The polar contribution is very low, and the contribution of the dispersive component to the free energy is more significant. This may be due to the many particles that generate induced dipoles produced in the charge distribution of the PHB chain. 

After 5 s of treatment, the contact angle of the distilled water decreased by 48%, while for diiodomethane there was an increase of 34%, and the SFE increased by 75%. This is due to the increase in the polar component since the dispersive component remains almost constant. This increase was associated with the functionalization on the surface of PHB with polar groups containing oxygen and hydrogen according to the Raman spectroscopy of the PHB coating, which is presented in the following section. 

In the samples treated at 10 s and 15 s, the dispersive component decreases slightly, which may be due to a restructuring in the charge distribution or polarizability induced by the APP in the electron cloud of the PHB. The polar component increases only by 3 mJ/mm^2^, resulting in slight increases in SFE. In the sample treated at 20 s, the dispersive component increases by 16%, and the polar component decreases by 19%; this could be related to the incision and degradation of the PHB chain.

### 3.4. X-ray Diffraction (XRD) of the PHB Coating

Figure 4 shows XRD patterns of the PHB coatings treated with APP for different plasma exposure times. It is shown in the diffractograms that there are no other diffraction peaks formed as a function of the treatment time. This indicates that the surface treatment with APP at different times does not lead to the appearance of other crystalline structures or phases.

The increase in the intensity of the crystalline peak at 13° shows that the treatment with APP produces an increase in the crystallinity degree of the PHB.

In Figure 5, the values for the percentage of the crystalline fraction of the PHB without treatment are presented. It is observed that after the different exposure times with APP, the PHB maintains a relatively low percentage of crystallinity compared to the sample without APP treatment. Because the chains are in an amorphous state, these polymeric macromolecules present a more significant space than the chains of the crystalline zones. This results in greater mobility in the chains when they are excited by the plasma, and a possible rearrangement of hydrogen bonds may occur between the hydrogen of a macromolecule and the oxygen linked by a double covalent bond to the carbon of another PHB chain. The restructuring mechanisms of the amorphous phase could be due to the interaction of the particles generated by the plasma and the PHB molecules, which may have reorganized the PHB structure.

A series of crystalline structures form crystals. In the case of polymers, the crystals can vary in size due to the distribution of the length of the polymer chains. The plasma generates incisions or breakages of the polymer chains; as there are reactive species, these macromolecules can grow or shorten by linking to other chains.

Figure 6 shows the size of the crystals formed at different treatment times. A decrease is observed within the first 5 s of APP treatment due to the intensity of the plasma within the initial seconds. This generates numerous chain breaks and initiates a crystal restructuring process. After 5 s, the crystals begin to grow again. This suggests that after 15 s of APP treatment, the chains begin to lengthen, causing the development of crystals.

### 3.5. Optical Characterization of PHB Coatings

Figure 7a shows the homogeneity of the PHB coatings from 0 s to 20 s of APP treatment. In Figure 7b, the red arrows indicate the swirl-shaped lumps caused by the rotation of the spin coating technique used to prepare the PHB coatings. There do not seem to be any significant morphological changes within the first 10 s of treatment, as shown in Figure 7c. However, in Figure 7d, when PHB is treated for 15 s, circling lumps appear on the surface (shown by blue circles), indicating that the plasma treatment is causing surface changes.

In the PHB coating treated with APP for 20 s (Figure 7e), the appearance of small lumps with a darker coloration is shown with red circles. This may indicate that the surface treatment begins to cause damage and possible localized degradation in different areas of the coating due to wear on the surface caused by ions.

### 3.6. Raman Spectroscopy of Coating Treated

Figure 8 shows the Raman spectra of the SS-PHB coatings for each treatment. The spectra reveal the vibration and rotation modes of the different links comprising the PHB chain. The vibration band of the C=O double bond appears at 1728 cm^−1^ in the crystalline phase, whereas it is absent at 1240 cm^−1^ in the amorphous phase.

This bond becomes more susceptible to polarizability after 5 s of treatment, decreasing after 10 and up to 20 s of PHB coating treatment with APP. The C-H stretch band of the CH3 group is most intense within the first 5 s of APP treatment, appearing at 2973 cm^−1^. It maintains a comparable relationship with the C=O double bond in terms of polarization changes induced by APP treatment.

Similarly, greater intensity in the Raman bands is seen at 3006 cm^−1^ in the first 5 s compared to those given by the energetic vibrations of the hydrogen bonds. The vibrations of the main skeleton, given by the C-O-C bonds of the helical conformation of the crystalline parts, can be observed at 1135, 1220, 1262, and 1298 cm^−1^ and undergo more changes within the first 5 s of PHB treatment [24]. 

The variation in intensities is related to the direction of molecular orientation in the plane of the direction of the PHB spherulite that changes orientation due to polarizability induced by APP treatment. Each band in the Raman spectrum corresponds directly to a specific vibrational frequency of a bond within the molecule. The vibrational frequency, and therefore the position of the Raman band, is susceptible to the orientation of the bands and the weight of the atoms at each end of the bond. More molecules forming crystals with the same orientation and strength results in a more intense band at a single frequency.

### 3.7. Potentiodynamic Polarization (PDP) Curves

Potentiodynamic polarization was employed to identify the kinetic properties related to SS corrosion in Hanks’ solution. Figure 9 displays the polarization curves of uncoated and coated SS treated at several times with APP, after a 1 h immersion in Hanks’ solution at 25 °C.

Table 2 presents the values of the corrosion potential (E_corr_), corrosion current density (i*_corr_*), cathodic (βc), and anodic (βa) Tafel slopes determined using the “Z view”. To determine the corrosion efficiency (ε_PDP_) from potentiodynamic polarization values, the following equation is used [25]:(3)$$\varepsilon_{PDP}=\frac{i_{corr}^{0}-i_{corr}^{c}}{i_{corr}^{0}}\times 100\%$$
where i^0^_corr_ and i^c^_corr_ denote the corrosion current densities of uncoated and coated treated with APP, respectively. 

Noticeably, the corrosion efficiency increased for coated samples treated at different times with APP, with the highest corrosion efficiency (91.14%) for the sample coated and treated for 20 min.

Remarkably, the corrosion current density in all polarization curves for PHB-coated samples treated with APP was approximately 85% lower than that of the uncoated SS sample. Additionally, i_corr_ decreased progressively with the increase in the treatment time (see Table 2). i_corr_ was reduced from 1.513 μA-cm^−2^ for the uncoated SS sample to 0.134 μA-cm^−2^ for the PHB-coated surface treated at 20 min, indicating a significant decrease in the corrosion rate on the PHB-coated surface.

The protective impact of the PHB coating caused a negative shift in corrosion potential (E_corr_). Especially as the treatment time of the PHB coating increased, the magnitude of the negative shift in corrosion potential became more pronounced, shifting from −257.3 mV for the uncoated sample to −283.3 mV for the PHB-coated sample treated for 20 min with APP. Furthermore, the PHB coating considerably reduced the corrosion current (91%).

Both Tafel slopes (βa and βc) of the PHB-coated surfaces treated have a lower value than that of the uncoated surface, with almost the same decrease in the cathodic and anodic Tafel slopes at different treatment times. This indicates that PHB-coated samples present corrosion via a mixed-type mechanism and suppression of anodic metal dissolution. 

Coating porosity is a valuable indicator in evaluating the appropriateness of a PHB coating treated with APP for protection. The porosity of these PHB coatings can be evaluated using [25]:(4)$$P=\frac{R_{p}(u)}{R_{p}(c)}\times 10^{-(\frac{\left|\Delta Ecorr\;\right|}{\beta a})}$$
where P represents the porosity, and Rp(u) (uncoated sample) and Rp(c) (PHB-coated and treated samples) indicate the polarization resistance. ΔE_corr_ is the difference in corrosion potentials between PHB-coated and uncoated samples, while βa is the anodic Tafel slope for the uncoated SS sample. The Rp(u) and Rp(c) polarization resistance values are calculated using the Stern–Geary equation [25]:(5)$$R_{p}=\frac{\beta_{a}\beta_{c}}{2.303\;i_{corr}\;(\beta_{a}{+\beta }_{c})}$$

The R_p_ values obtained are presented in Table 2. As can be seen, the R_p_ values for the PHB-coated sample and those treated with APP increase by one order of magnitude in comparison with those of the uncoated sample. This signifies a greater capacity to resist corrosion, indicating an improved protective effect. The R_p_ increase, together with the decrease in the corrosion current densities, means a substantial enhancement of the corrosion resistance for PHB-coated samples and those treated with APP. 

On the other hand, the porosity (P) decreases with the increase in the APP treatment time. This implies that the treatment with APP tends to deposit PHB coating on the surface sample with better uniformity at higher treatment times. 

Based on the present results, it can be determined that at higher APP treatment times, the PHB coating on the SS sample exhibits less porosity. This decrease in porosity is linked to a reduction in corrosion current density (i_corr_) and an increase in protection efficiency (ε_PDP_). Therefore, the APP treatment time of coated samples ensuring the optimal corrosion resistance is 20 min.

### 3.8. Electrochemical Impedance Spectroscopy and Bode Plots

EIS was employed to assess the corrosive properties and obtain information on the electrochemical processes at the solution–electrode interface of the PHB coatings treated with APP. 

Figure 10 shows Nyquist and Bode plots of EIS of PHB-coated samples treated at several times after a 1 h immersion in Hanks’ solution. Nyquist plots (Figure 10a) display a single semicircle at lower frequencies, indicating resistance to the charge transfer from the coating to the electrolyte.

Figure 10b displays Bode plots of the SS-PHB-coated samples. A significant increase in the total impedance modulus (|Z|) of PHB-coated samples in the lower frequency region suggests good protection of the SS samples. 

On the other hand, in the Bode plot, the middle frequency range displays an angle of approximately 50°, which is identified as capacitive behavior. Likewise, the phase maximum associated with the double electric layer increases with an increase in the treatment time, implying a reduction in the corrosion rate of the electrode. Also, it can be observed that the semicircle increases as a function of the treatment time, with the highest impedance for the PHB-coated sample at 10 s. This implies a strong enhancement of corrosion resistance due to the protection given by the PHB coating.

### 3.9. Equivalent Circuits

Figure 10a depicts the equivalent electrical circuit model that was fitted to the EIS data. Here, Rs indicates the solution resistance; Ra denotes the resistance of the PHB-coated film, depending on the adsorption layer; Rct represents the charge transfer resistance; and Ca and Cdl are constant phase elements that define the non-ideal capacitance behavior of the adsorption film and double electric layer, respectively. Wd represents the Warburg diffusion impedance.

Table 3 shows that the Rct value reaches its maximum value of 6370 Ω·cm^2^ at a treatment time of 10 s, higher than that of the sample without PHB coating. This is attributed to the effective barrier behavior of the PHB coating; in other words, it suggests an improved protective barrier capacity against the electrolyte and remarkable corrosion protection of the steel surface against Hanks’ solution. This confirms the results obtained from potentiodynamic polarization tests.

On the other hand, the capacitance behavior of Cdl related to the double layer, which decreases with the increase in treatment time, indicates a decrease in the corrosion rate of the electrode. The Warburg diffusion impedance (Wd) of PHB-coated samples treated with APP increases considerably compared to that of the untreated sample, suggesting a reduction in diffusion from the electrolyte through the finite thickness of the PHB coating, resulting in a transmissive (porous) boundary.

### 3.10. Scanning Electron Microscopy After Electrochemical Tests

In Figure 11a, the images obtained with scanning electron microscopy show that the base steel exhibits areas with pitting corrosion, characteristic of the environment to which it is subjected.

In Figure 11b, the PHB coating with varying thicknesses is observed. This variation in thickness can lead to anodic and cathodic reactions due to differences in homogeneity [26]. Additionally, white spots are visible; these may result from the deposition of salts or phosphates from the solution, leading to species diffusion within the PHB coating and inducing the degradation effect via the crack formation on the PHB coating. This phenomenon is more pronounced in denser and transparent areas of the coating where there is poor adhesion between the steel and the PHB. In Figure 11c, salt deposition can be observed, but the PHB coating is peeling slightly due to the diffusion of species originating from the electrolyte. This leads to the swelling and peeling of the PHB coating from the steel–PHB interface. In Figure 11d, the cracking of the polymeric film is observed with slight deposits of salts, noticeable as dispersed white spots on the surface. These deposits contain ions from the solution, and the cracking is induced by ion exchange from the solution into the coating. This transport involves substances denser than electrons, which break the polymeric chains of the coating or induce chemical attack. Figure 11e shows and confirms only the salt deposits on the film, as well as exhibiting slight swelling in less dense areas. In Figure 11f, the propagation of a crack and slight white spots within of them are observed, indicating areas more susceptible to damage [27].

### 3.11. Raman Spectroscopy After Electrochemical Tests

Figure 12 shows the Raman spectra measured after performing the electrochemical tests on the PHB coatings treated with APP; a visible degradation of the PHB coating is observed in all the Raman spectra.

Table 4 presents the bands that suffer degradation in the Hanks’ solution. This degradation arises from the charge exchange occurring during the reduction and oxidation processes in the PHB–electrolyte interface. The present results confirm the degradation observed in the electrochemical test due to the species change from the electrolyte to the PHB coating.

## 4. Conclusions

The principal results of the present investigation regarding the improvement of PHB coatings on SS are as follows:-A film of approximately 15 ± 1 μm thickness was successfully synthesized on SS using the spin coating technique.-Surface treatment with APP on 316L steel increases the surface free energy but does not promote the adhesion of PHB coatings, as verified through contact angle measurements.-Superficial treatment of the PHB coating without prior APP treatment of the steel significantly enhances polymer adhesion. This results in an increase in the surface free energy of the PHB coating, confirmed through pull-off adhesion tests.-The PHB coating treated with APP undergoes an increase in crystallinity (confirmed through XRD).-APP treatment induces high polarizability in the PHB coating, facilitating the insertion of functional groups and the formation of hydrogen bonds that can aid in drug binding (demonstrated through Raman spectroscopy).-The PHB coating reduces the corrosion current in steel, making it a promising biopolymer for controlling the corrosion rate of alloys used in medical applications.-The results from potentiodynamic polarization and EIS investigations indicate that the PHB coating forms a highly effective anti-corrosion layer on SS in Hanks’ solution. This underscores the crucial role of coating properties in the protective performance of SS. The best protection against corrosion was observed when the PHB coating was treated for 20 s.

These conclusions highlight the potential of PHB coatings for enhancing corrosion resistance in medical applications.

## Figures and Tables

**Figure 1 polymers-16-02073-f001:**
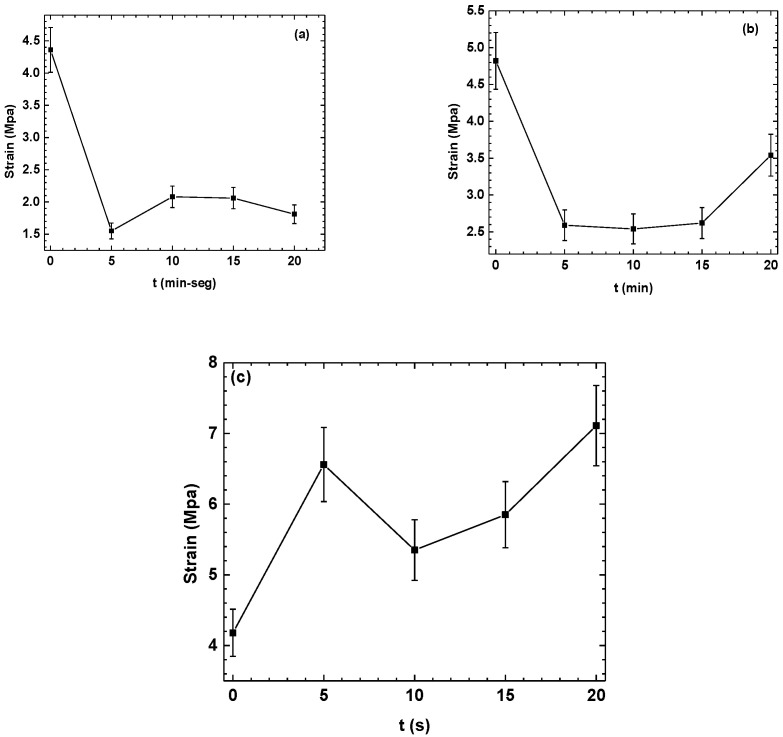
Adhesion tests. Methodology: (**a**) M1; (**b**) M2; (**c**) SS-PBH.

**Figure 2 polymers-16-02073-f002:**
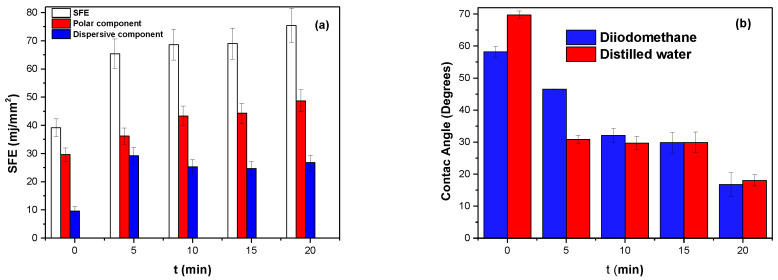
(**a**) SFE as a function of treatment time, (**b**) contact angles on steel as function of treatment time.

**Figure 3 polymers-16-02073-f003:**
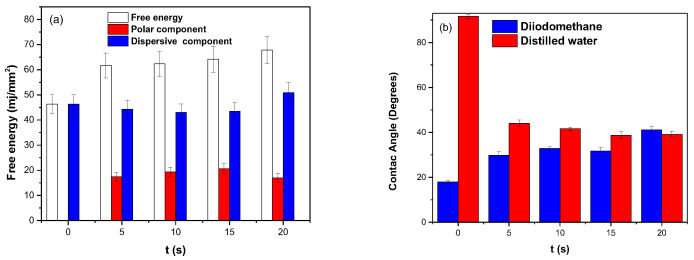
(**a**) SFE as a function of treatment time, (**b**) contact angles on polymer as a function of treatment time.

**Figure 4 polymers-16-02073-f004:**
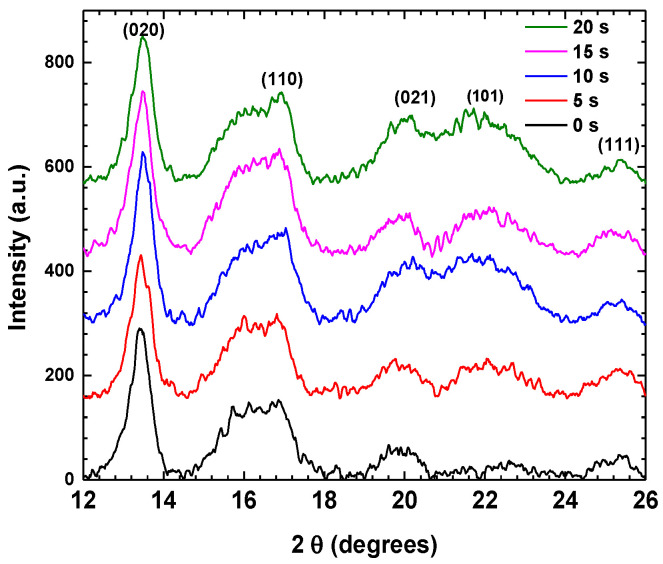
XRD spectra of PHB coating treated with APP.

**Figure 5 polymers-16-02073-f005:**
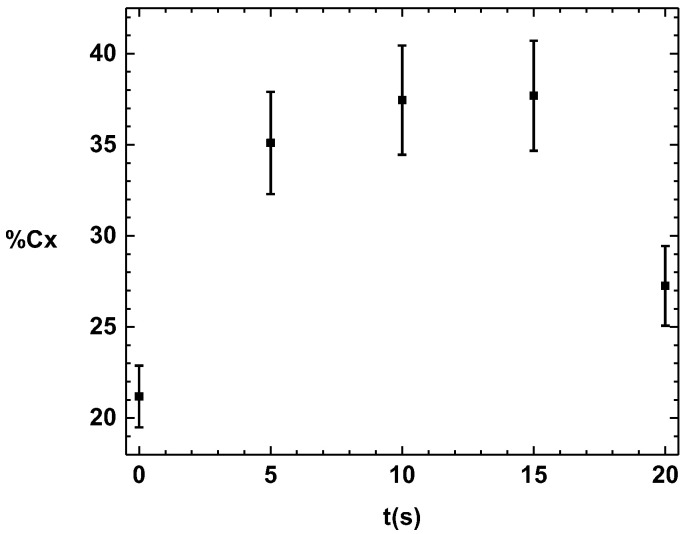
Percentage of crystallinity of coating as a function of treatment time.

**Figure 6 polymers-16-02073-f006:**
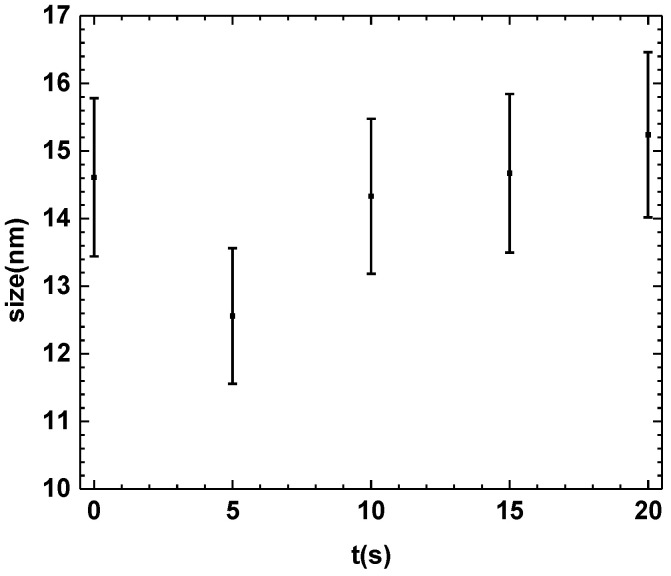
Crystal size at different treatment times.

**Figure 7 polymers-16-02073-f007:**
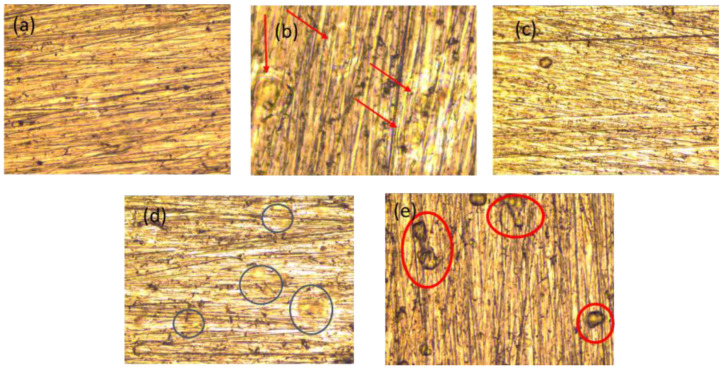
Optical characterization of the treated coating at different times:(**a**) 0 s, (**b**) 5 s, (**c**) 10 s, (**d**) 15 s, (**e**) 20 s.

**Figure 8 polymers-16-02073-f008:**
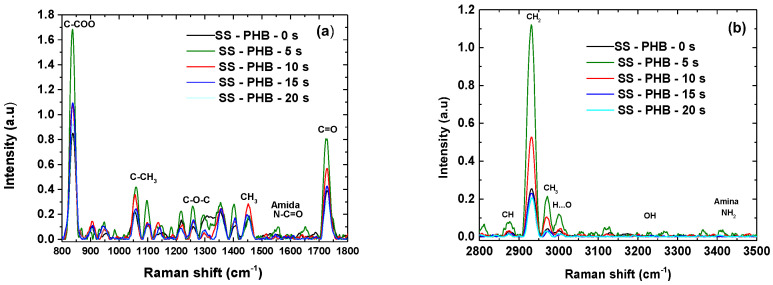
Raman spectra of SS-PHB coatings at different times. (**a**) Region of 800 to 1800 cm^–1^; (**b**) region of 2800 to3500 cm^−1^.

**Figure 9 polymers-16-02073-f009:**
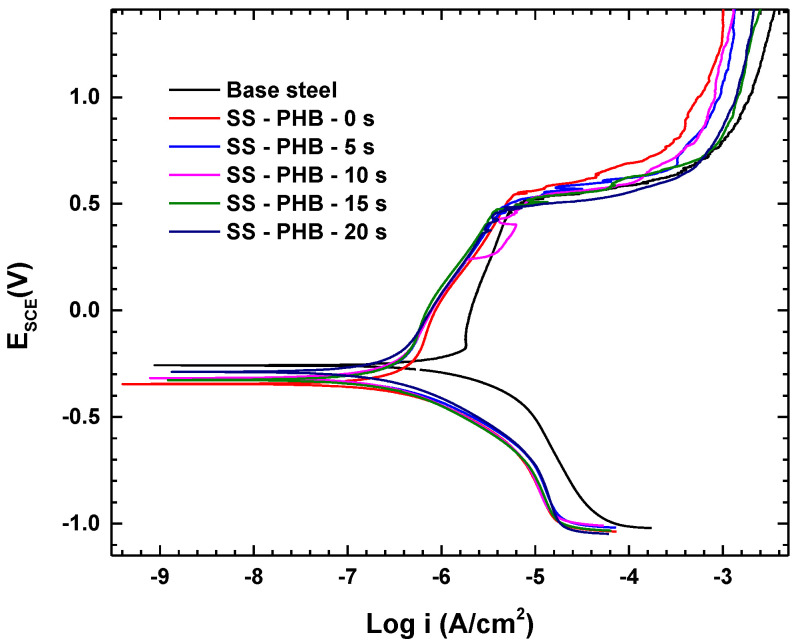
Polarization curves.

**Figure 10 polymers-16-02073-f010:**
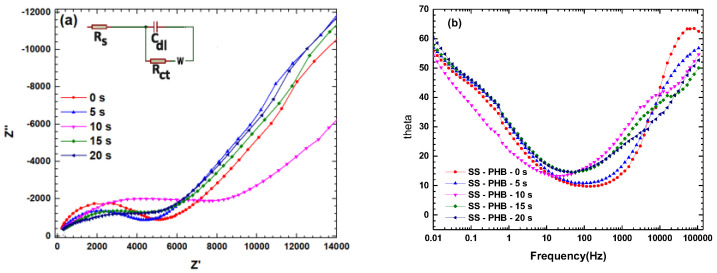
(**a**) Nyquist diagram, (**b**) Bode diagram.

**Figure 11 polymers-16-02073-f011:**
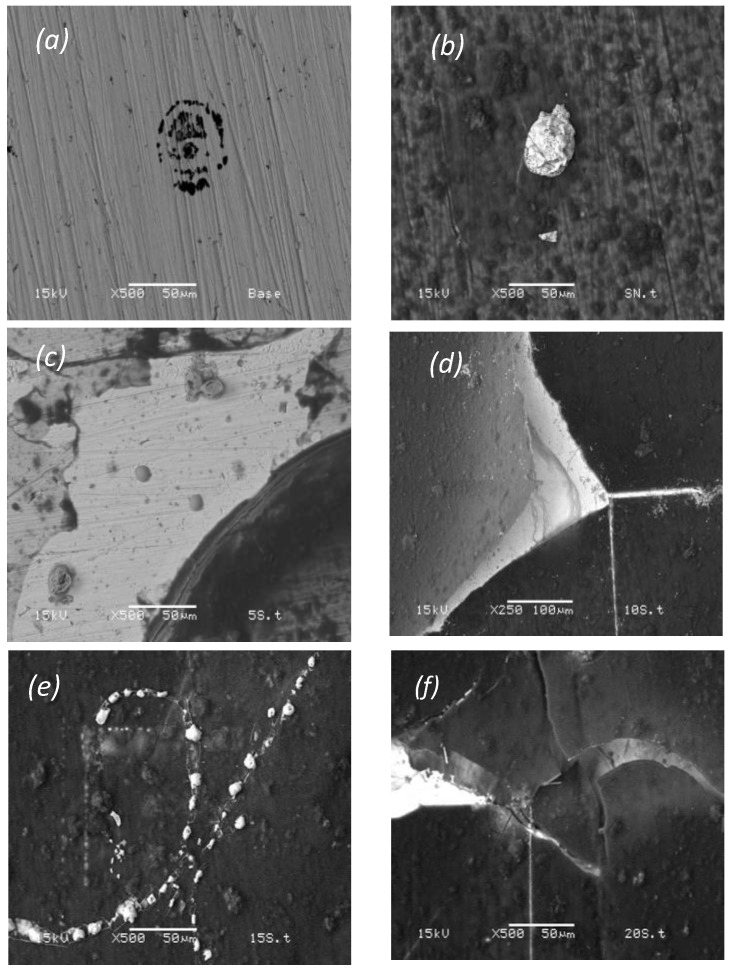
Scanning electron microscopy: base steel (**a**), coated SS (**b**), coated SS treated at 5 s (**c**), coated SS treated at 10 s (**d**), coated SS treated at 15 s (**e**), coated SS treated at 20 s (**f**).

**Figure 12 polymers-16-02073-f012:**
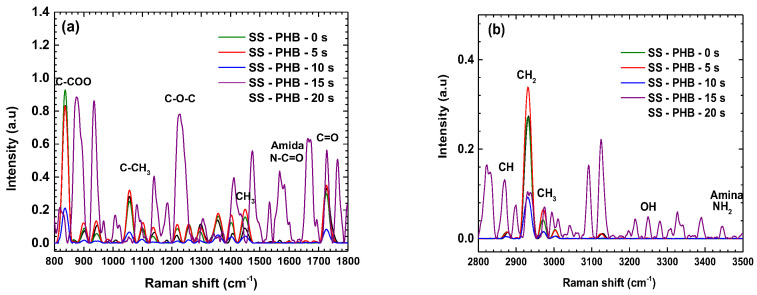
Raman spectra of PHB after electrochemical tests: (**a**) from 800 to 1800 cm^−1^, (**b**) from 2800 to 3500 cm^−1^.

**Table 1 polymers-16-02073-t001:** Treatment times for the different methodologies.

M1		M2	SS-PHB
t (min)	t (s)	T (min)	t (s)
0	0	0	0
5	5	5	5
10	10	10	10
15	15	15	15
20	20	20	20

**Table 2 polymers-16-02073-t002:** Potential and current corrosion values.

	*E_corr_* (mV)	*I_corr_ *(µA/cm^2^)	*β_a_* (mV/dec)	*β_c_* (mV/dec)	$\varepsilon_{PDP}$ (%)	R_p_ (%)	P (%)
SS Substrate	−257.3	1.513	1106.4	239.3		56.46	
S/T	−345.1	0.234	740.6	226.1	84.53	321.43	17.56
5 min	−321.0	0.199	677.0	217.7	86.85	359.44	15.71
10 min	−317.5	0.147	780.7	221.3	90.28	509.32	11.09
15 min	−326.9	0.158	895.3	236.9	89.56	514.82	10.97
20 min	−283.3	0.134	742.3	222.5	91.14	554.72	10.18

**Table 3 polymers-16-02073-t003:** Numerical values of the elements for the equivalent circuits of substrate steel without and with PHB coatings.

Condition	R_1_ (Ω-cm^2^)	R_2_ (Ω-cm^2^)	C_dl_ (µF/cm^2^)	W_R_ (Ω-cm^2^)	W_c_ (F/cm^2^)	Xi^2^
**0 s**	84.01	3819	0.02	7145	0.09	2.0 × 10^−4^
**5 s**	37.57	3548	0.10	5020	0.11	1.0 × 10^−4^
**10 s**	15.30	6370	0.74	9501	0.36	9.0 × 10^−4^
**15 s**	8.43	4433	0.01	4842	0.06	8.0 × 10^−4^
**20 s**	4.29	3308	0.69	8146	0.26	1.5 × 10^−3^

**Table 4 polymers-16-02073-t004:** Percentages of degradation after the electrochemical tests.

Band	Raman Shift (cm^−1^)	Intensity (a.u.)	% Reduction	Band	Raman Shift (cm^−1^)	Intensity (a.u.)	% Reduction
C-COO	835	0.91	54.49	CH	2878	0.01	14.29
C-CH_3_	1060	0.25	60.98	CH_2_	2931	0.27	24.11
O-C-O	1101	0.1	47.62	CH_3_	2973	0.03	14.29
CH_3_	1453	0.14	82.35	O..H	3003	0.02	18.18
N-C=O	1555	0.01	10.31	OH	3225	0.01	50.00
C=O	1728	0.31	39.24	NH_2_	3357	0.01	33.33

## Data Availability

Data is contained within the article.

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
