# Peer review of "Atmospheric Plasma Treatment to Improve PHB Coatings on 316L Stainless Steel"

_polymers, 2024, doi:10.3390/polym16142073_

Round 1

Reviewer 1 Report

Comments and Suggestions for Authors

The article presents a large amount of experimental material and analyzes the works of many researchers. However, there are the following remarks:

1. Section 2.1 it is necessary to justify why these particular modes of preparation of steel and polymers were chosen.

2. What do the authors mean by the phrase "The APP was characterized and reported previously." (lines 93-94). Where is the APP process characterized and described? If the author means lines 62-67, this is a very brief citation of the literature without characterizing the process and describing it.

3. Lines 99-108 are shifted to a table in my version of the paper. The title of section 2.3 also needs to be corrected. The same lines 294-295.

4. Figure 8: in the caption, a and b must be deciphered.

5. Figure 12: some findings are questionable, more detail is needed. For example: Figure 12(d) presents a more homogeneous surface with less degradation and exhibits small salt deposits indicated by white discoloration. Figures 12(e) and 12(f) show a more homogeneous structure. There is no description of figure 12(f).

6. How was the film thickness adjusted and evaluated? Line 531-532: A film of approximately 15 μm thickness was successfully synthesized on SS using the spin coating technique.

Author Response

Thank you for your review and comments of our manuscript. Your evaluation of us is very pertinent and the questions you have asked are very precise. We have carefully analyzed your questions and will respond to each of them below.

  1. Section 2.1 it is necessary to justify why these particular modes of preparation of steel and polymers were chosen.

We added a paragraph:

To determine the best procedure to coat SS with PHB using APP, we use the formula for combining 3 elements (SS, PHB, APP) to get the best coating, that is That gives three different methodologies for the …

  1. What do the authors mean by the phrase "The APP was characterized and reported previously." (lines 93-94). Where is the APP process characterized and described? If the author means lines 62-67, this is a very brief citation of the literature without characterizing the process and describing it.

The paragraph “The APP was characterized and reported previously. The plasma conditions included an electron density of (1.95 ± 0.24) × 1012 cm−3 and an electron energy of (0.36 ± 0.04) eV, which were obtained through electrical and optical characterization, respectively [19].”

Was replaced by

With the present condition the electric field strength (E=V/d) was 5.0 × 103 V cm-1, E/P = 6.58 V cm-1 -torr-1, an electron drift vd = 3.2 × 106 cm-s-1 and E/N of 0.19 Td. That gives an electron density of (1.95 ± 0.24) × 1012 cm-3[19]. The APP was also characterized by optical emission spectroscopy from 200 to 900 nm, the lines and bands observed were the OH(A2Σ+ – X2Πi); N2(C3Πu → B3Πg); N2+(B2Σu+ → X2Σg+); O (5S05P) and O (3S03P). The electron energy of the plasma was found using an optical characterization of the discharge, from the ratios of the O atomic line intensities in the emission spectrum. The electron energy obtained was (0.36 ± 0.04) eV [19].

  1. Lines 99-108 are shifted to a table in my version of the paper. The title of section 2.3 also needs to be corrected. The same lines 294-295.

They have been corrected.

  1. Figure 8: in the caption, a and b must be deciphered.

We added to the caption of figure 8:

 (a) Region of 800 to 1800 cm-1; (b) Region of 2800 to3500 cm-1.

  1. Figure 12: some findings are questionable, more detail is needed. For example: Figure 12(d) presents a more homogeneous surface with less degradation and exhibits small salt deposits indicated by white discoloration. Figures 12(e) and 12(f) show a more homogeneous structure. There is no description of figure 12(f).

Thank you very much for point out that discussion.

We rewrote that part also change Figure 12 to Figure 11, due to Figure 11 no exist in the firs version::

In Figure 12(d), the cracking of the polymeric film is observed with slight deposits of salts, noticeable as dispersed white spots on the surface. These deposits contain ions from the solution, and the cracking is induced by ion exchange from the solution into the coating. This transport involves substances denser than electrons, which break the polymeric chains of the coating or induce chemical attack. Figure 12(e) shows and confirms only the salt deposits on the film, as well as exhibiting slight swelling in less dense areas. In Figure 12(f), the propagation of a crack and slight white spots within of them are observed, indicating areas more susceptible to damage.

  1. How was the film thickness adjusted and evaluated? Line 531-532: A film of approximately 15 µm thickness was successfully synthesized on SS using the spin coating technique.

The film thickness was adjusted as a function of the volume deposited (3 ml for each sample) and a deposition time of 1 min. So, we added the following paragraph in section 2.1:

The P3HB solution (3 mL for each sample) was injected onto the SS at 300 rpm for 1 min using a spin coater. The thickness of the coatings was measured in triplicate using a WEN brand digital micrometer. The film thickness obtained was 15 ± 1 µm.

Reviewer 2 Report

Comments and Suggestions for Authors A magnificent work on coating 316L steel samples with biopolymers. Only two questions that I would like the authors to clarify for me, although the work can be published in its current form. 1. In figure 7 I see something similar to "preferential orientation", is it an intended effect? 2. An experimental study of the possible aging of the samples is missing. Is it running?

Author Response

A magnificent work on coating 316L steel samples with biopolymers. Only two questions that I would like the authors to clarify for me, although the work can be published in its current form.

Thank you for your recognition of our manuscript. We have carefully analyzed your questions and will respond to each of them below.

  1. In figure 7 I see something similar to "preferential orientation", is it an intended effect?

You are correct, we made the samples with preferential orientation to avoid any difference between them to have a clear comparation of the experimental condition and better results.

  1. An experimental study of the possible aging of the samples is missing. Is it running?

Thank you very much for point out, the experimental study of the aging of the samples is running.  We believe that results could be included in future work.